# Personalized Facial Expressions and Head Poses For Speech-Driven 3D Facial Animation

## Abstract

Speech-driven 3D facial animation aims at generating facial movements that are synchronized with the driving speech, which has been widely explored recently. Existing works mostly neglect the person-specific talking style in generation, including facial expression and head pose styles. Several works intend to capture the personalities by fine-tuning modules. However, limited training data leads to the lack of vividness. In this work, we propose **AdaMesh**, a novel adaptive speech-driven facial animation approach, which learns the personalized talking style from a reference video of about 10 seconds and generates vivid facial expressions and head poses. Specifically, we propose mixture-of-low-rank adaptation (MoLoRA) to fine-tune the expression adapter, which efficiently captures the facial expression style. For the personalized pose style, we propose a pose adapter by building a discrete pose prior and retrieving the appropriate style embedding with a semantic-aware pose style matrix without fine-tuning. Extensive experimental results show that our approach outperforms state-of-the-art methods, preserves the talking style in the reference video, and generates vivid facial animation. The supplementary video and code will be available at https://adamesh.github.io.

## 1 Introduction

Speech-driven 3D facial animation has been widely explored and attracted increasing interest from both academics and industry. This technology has great potential in virtual reality, film production, and game creation. Most previous works focus on improving synchronization between speech and lip movements (Cudeiro et al., 2019; Richard et al., 2021; Fan et al., 2022; Xing et al., 2023), however, they have neglected the personalized talking style, including the facial expressions and head poses. The facial animation with expressive facial expressions and diverse head poses contributes to a more vivid virtual character and grants users with a better interaction experience.

Recently, some works (Ye et al., 2023; Thambiraja et al., 2023) attempt to model the person-specific talking style by fine-tuning or adapting modules. In the real application, only a limited amount of video clips, even shorter than 1 minute, are provided by target users to capture the personalized talking style. These methods thus meet several challenges. 1) The few adaptation data probably induces catastrophic forgetting (Zhai et al., 2023) for the pre-trained model and easily leads to the overfitting problem (Zhang et al., 2022). Specifically for the facial expressions, after the adaptation, the lip synchronization and richness of expressions significantly decrease for the unseen speech inputs. 2) Speech is a weak control signal for the head poses (Henter et al., 2020; Zhang et al., 2021). Adaption with few data or learning mappings on such weak signals leads to averaged generation. The predicted head poses are in lack of diversity and have a smaller variation of movements.

In this work, to address these two challenges, we propose an adaptive speech-driven 3D facial animation approach, AdaMesh, consisting of an expression adapter and a pose adapter. To tackle the catastrophic forgetting and overfitting problem, we devise an expression adapter built upon the low-rank adaptation technique, LoRA(Hu et al., 2022), for its high data efficiency in few-shot model adaptation. Since the facial expressions contain temporal dynamics, we propose a novel mixture-of-LoRA (MoLoRA) strategy, featured by employing LoRA with different rank sizes to model the multi-scale dynamic facial details.

As for the averaged pose generation problem, we recognize it as a repercussion of generating poses via motion regression (Xing et al., 2023) and a mismatch of the adapted pose style. Hence we formu-

late the pose generation task as a generative one, for which we employ a vector quantized-variational autoencoder (VQ-VAE) (van den Oord et al., 2018) and a generative pre-trained Transformer (Pose GPT) network (Vaswani et al., 2017) for the backbone of the pose adapter. The head poses are mostly associated with the conveyance of negation, affirmation, and turnaround in semantics. To consider and balance both strong and weak semantic associations in head pose patterns, we propose the retrieval strategy for the pose style adaptation with a novel semantic-aware pose style matrix.

The main contributions of our work can be summarized as follows:

- We propose AdaMesh, which is the first work that takes both facial expressions and head poses into account in personalized speech-driven 3D facial animation.
- We are the first to introduce LoRA to the 3D facial animation task and propose MoLoRA to efficiently capture the multi-scale temporal dynamics in facial expressions from limited data.
- We propose a pose adapter, featured by a semantic-aware pose style matrix and a simple retrieval strategy for more diverse pose generation without fine-tuning any parameters.
- We conduct extensive quantitative, qualitative and analysis on our AdaMesh. Results show that AdaMesh outperforms other state-of-the-art methods and generates a more vivid virtual character with rich facial expressions and diverse head poses.

## 2 RELATED WORK

**Speech-Driven 3D facial Animation.** The recent speech-driven 3D facial animation methods can be divided into two categories according to the format of animation. The first category of methods (Cudeiro et al., 2019; Richard et al., 2021; Fan et al., 2022; Xing et al., 2023; Thambiraja et al., 2023) directly predicts vertices of the face from speech and renders the vertices to meshes for visualization. The vertice data is collected through professional 3D scans of human heads (Fanelli et al., 2010; Cudeiro et al., 2019; Li et al., 2017), which is expensive and time-consuming. The scanned data precisely describes the facial movements. The methods trained on such data usually have authentic lip movements. Another category of methods predicts parameters of expressions (Daněček et al., 2023) or blendshapes (Chen et al., 2022; Peng et al., 2023). These parameters can be reconstructed from the scanned vertices or 2D face videos through a parametric face model (Li et al., 2017; Peng et al., 2023). However, the decreasing quality caused by the reconstruction usually leads to lower speech-lip synchronization. The latest methods that take personalized style into consideration mostly adopt expression parameters for style modeling, since the scanned vertices are in the neutral style and lack rich expressions. Emotalk (Peng et al., 2023) and EMOTE (Daněček et al., 2023) are related works, which disentangle content and emotion to generate emotional talking style. But they adopt private or unavailable reconstruction pipelines. The most related work is Imitator (Thambiraja et al., 2023), which generates meshes and leverages the expressions from 2D video for style adaptation.

**Head Poses Generation.** To the best of our knowledge, existing speech-driven 3D facial animation methods neglect the head poses, since the scanned vertices have no head movements. Some portrait-realistic talking face generation methods attempt to model the head poses. A typical way is to copy the head pose sequence from a real video recording (Thies et al., 2020; Ye et al., 2023; Chen et al., 2023), which can definitely generate realistic head poses. However, these methods disregard the correlation between speech and head pose, and the repetitive and non-engaging head poses create unnaturalness during long periods of interaction. The rapidly growing research topic, gesture generation from speech (Zhu et al., 2023), can also justify the importance of predicting head poses from speech rather than utilizing a template pose sequence. Another way is to predict head poses from speech (Lu et al., 2021; Zhang et al., 2021). The most effective method for head poses is FACIAL (Zhang et al., 2021), which proposes an implicit attribute learning framework to generate natural head poses. However, FACIAL sometimes generates static head poses and lacks pose diversity. Moreover, these methods can only generate head poses for a specific person, and cannot realize flexible style adaptation. Our approach could be regarded as bridging the advantages of the aforementioned two ways. The quantized pose space learned by VQ-VAE ensures realism, while the prediction step outputs natural but not repetitive pose sequence.

**Adaptation Strategy.** Efficient adaptation strategies of pre-trained models, including adapter-based (Rebuffi et al., 2017), prefix-based (Li & Liang, 2021), and LoRA (Hu et al., 2022) techniques, have been applied to various downstream applications. These strategies achieve advanced performance

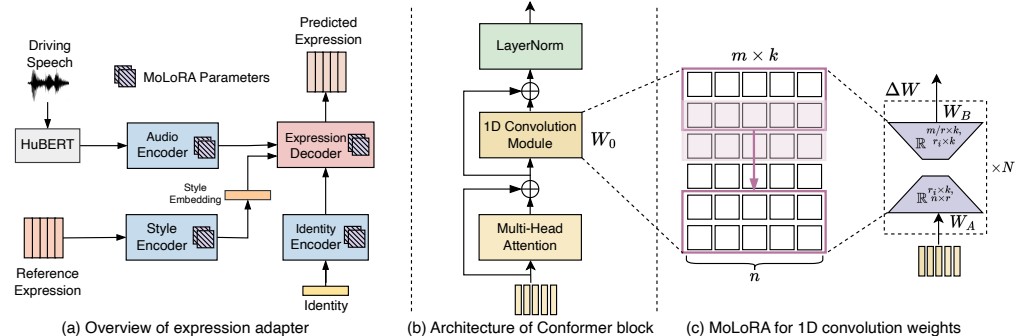

Figure 1: The overview of expression adapter. (a) The patches on each encoder and decoder denote the MoLoRA parameters which are added to the pre-trained modules after adaptation. (b) A brief illustration of Conformer (Li et al., 2021) (c) demonstrates that MoLoRA is applied to the convolution operator. It is also applied to the linear layers in the proposed expression adapter.

without fine-tuning all the model parameters. In the field of image synthesis (Rombach et al., 2022), LoRA serves as a plug-and-play extension for the pre-trained model to bring tremendous and specific styles to the images. This work is inspired by this idea and intends to bring the person-specific style to the pre-trained networks that predict facial expressions from speech. A similar idea in speech synthesis (Hsieh et al., 2022) manages to clone the voice of a new speaker with LoRA. Note that we refer to the models proposed in Sec. 3.1 and Sec. 3.2 as the adapters. It is different from the concept of adapter-based strategy Rebuffi et al. (2017) mentioned above.

**Remark.** Our work differs from previous works by the enhanced data efficiency in style adaptation and overcomes the challenges of catastrophic forgetting for facial expressions and averaged pose generation. To the best of our knowledge, we are the first to simultaneously take facial expressions and head poses into account in speech-driven 3D facial animation.

## 3 ADAMESH

In this section, we propose AdaMesh, a data-efficient speech-driven 3D facial animation approach tailored for generating personalized facial expressions and head poses. Given the extracted facial expression sequence and head pose sequence from a reference video, together with a driving speech, the expression adapter (Sec. 3.1) is adapted with a specific expression style and outputs rich facial expressions, while the pose adapter (Sec. 3.2) derives a semantic-pose aware style matrix and outputs diverse head poses. The generated facial expressions and head poses are combined by a parametric face model (Li et al., 2017) and rendered to mesh images.

### 3.1 EXPRESSION ADAPTER

To achieve efficient adaptation for facial expressions, we pre-train the expression adapter to learn general and person-agnostic information that ensures lip synchronization and then optimize the MoLoRA parameters to equip the expression adapter with a specific expression style. As shown in Fig. 1 The expression adapter is composed of an audio encoder, an identity encoder, a style encoder, an expression decoder, and MoLoRA to fine-tune convolutional and linear layers in these modules.

**Audio encoder.** To obtain robust semantic information from speech, we leverage HuBERT (Hsu et al., 2021), a self-supervised speech representation model, to extract features from raw speech. Unlike Wav2Vec (Baevski et al., 2020) used in previous methods (Fan et al., 2022; Xing et al., 2023), HuBERT can preserve more semantic information and provide better cross-lingual ability. The speech features are downsampled to the same length with expression sequence using convolutional layers with stride 2 and then sent into three Conformer layers (Li et al., 2021).

**Identity encoder.** Previous methods usually represent the identity with the one-hot label (Xing et al., 2023; Thambiraja et al., 2023) in the animation prediction. They show bad generalization on

unseen speaker identity. We utilize the identity parameter extracted by the parametric face model (Li et al., 2017; Feng et al., 2021). The identity sequence in a data sample is averaged along the time dimension to represent the speaker identity and then sent into three convolutional layers with conditional layer normalization (Chen et al., 2021).

**Style encoder and Expression Decoder.** To provide the talking style which cannot be directly predicted from the semantic speech information, the expression sequence is utilized as a residual condition to the expression prediction. The sequence is compressed to a compact style embedding after two Conformer layers and then concatenated with the encoded speech and identity features into the decoder. The decoder also consists of three Conformer layers, which are stacked in a residual manner rather than cascadation. It is inspired by the residual vector quantization (Zeghidour et al., 2021), which can realize coarse-to-fine prediction.

**MoLoRA.** We focus on the low-rank training techniques and propose MoLoRA to equip the expression adapter with the inductive bias of new data. As it is known, the vanilla LoRA (Hu et al., 2022) injects trainable rank decomposition matrices into each linear layer of the Transformer architecture. Since the convolutional layer can be viewed as multiple linear transformations with sliding receptive windows, we extend LoRA to the convolution operator for more flexible trainable parameters. The facial expressions contain multi-scale temporal dynamics, where the coarse-grained expression style (*e.g.,* emotion) and fine-grained facial details (*e.g.,* speech-sync lip movements, muscle random wrinkles) inherent in the expression sequence. We empirically found that the vanilla LoRA with only a single rank size filters out the multi-scale dynamics and tends to produce over-smoothed facial expressions. Hence we propose MoLoRA to mix $N$ LoRA with different rank sizes $r_i$. Supposing the pretrained weight matrix of 1D convolution $W_0 \in \mathbb{R}^{m \times n \times k}$, we define the low-rank product $\Delta W^i = W_B^i W_A^i$, where $W_B^i \in \mathbb{R}^{m/r_i \times k, r_i \times k}$ and $W_A^i \in \mathbb{R}^{r_i \times k, n \times r_i}$. The weights $W$ after adaptation merge the pre-trained and MoLoRA weights, which can be defined as:

$$W = W_0 + \Delta W = W_0 + \sum_{i=0}^{N} W_B^i W_A^i. \tag{1}$$

There is a concurrent work (Lialin et al., 2023) that investigates the efficiency in LoRA. However, it stacks multiple LoRA of a single rank size, while MoLoRA focuses on modeling the multi-scale characteristic in facial expressions.

**Procedure of the expression adapter.** The pretrain, adaptation, and inference procedures of the expression adapter are listed in Alg. 1. Note that the inference stage has a slight difference with the training. The style embedding is obtained from another expression sequence rather than the ground truth. However, this mismatch has no impact on the generation, since the style embedding learns a global talking style for the expression sequence rather than the semantic speech information.

---

**Algorithm 1** Pretrain, Adaptation, and Inference of Expression Adapter.

1: **Pretrain:** Train the Expression Adapter $\theta_E$ with source training data. The expression sequence for the style encoder is the same with the prediction of the expression adapter.
2: **Adaptation:** Update all low-rank matrices $\theta_{MoLoRA}$ for convolution and linear layers in the expression adapter with the adaptation data.
3: **Inference:** Merge $\theta_E$ and $\theta_{MoLoRA}$. Given a driving speech, and using an expression sequence from the adaptation data as the input for the style encoder, the expression adapter generates expressions that synchronize with the speech and reenact the facial style of the adaptation data.

---

### 3.2 Pose Adapter

To overcome the averaged generation problem in pose style adaptation, we propose the pose adapter and formulate the adaptation as a simple but efficient retrieval task instead of fine-tuning modules. As illustrated in Fig. 2, we first learn a discrete latent codebook $\mathcal{Z} = \{z_k \in \mathbb{R}^{d_z}\}_{k=1}^M$ of head poses with the novel sequence-encoding VQ-VAE (Siyao et al., 2022), and predicts the speech-relevant head poses with Pose GPT. The VQ-VAE training requires no speech input, so various video datasets of a larger amount can be utilized. This design enables the coverage of a wide range of poses, which is well-suited for learning a realistic pose manifold (Ng et al., 2022). A semantic-aware pose style matrix is derived for each pose sample in the training sample to create a database. We retrieve the

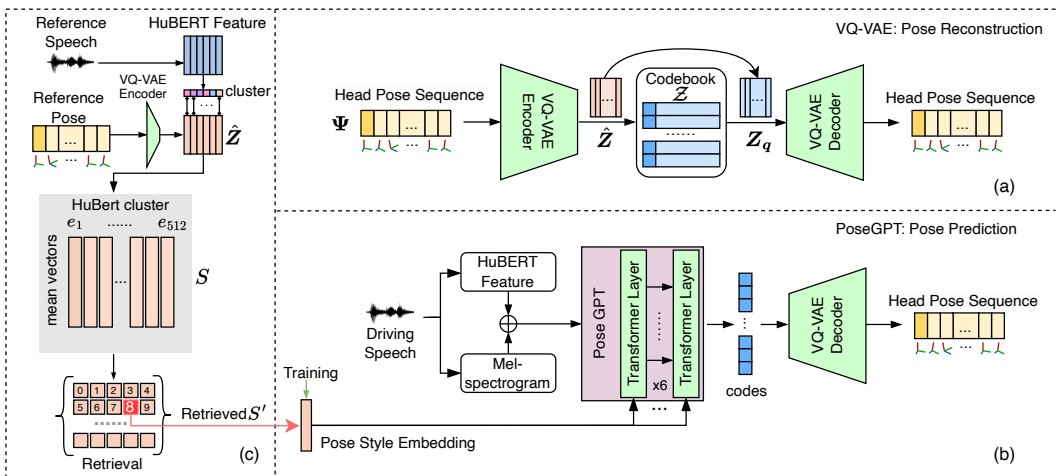

Figure 2: The overview of pose adapter. (a) Training of VQ-VAE. (b) Pose GPT. (c) The derivation of the semantic-aware pose style matrix and the retrieval strtegy. In the training of the Pose GPT, the pose style embedding is the assigned one-hot label. $S$ denotes the semantic-aware pose style matrix.

nearest matrix in the pose style matrix database as the adapted style matrix in the adaptation stage. The simple retrieval strategy for adaptation is based on the assumption that the possible pose styles are covered in the VQ-VAE learning with the large-scale dataset.

**Discrete Pose Space.** Given a head pose sequence $\boldsymbol{\Psi} = [\boldsymbol{\psi}_1, \dots, \boldsymbol{\psi}_T] \in \mathbb{R}^{T \times 3}$, the VQ-VAE encoder, which consists of 1D convolutional layers, firstly encodes it into context-aware features $\hat{\boldsymbol{Z}} \in \mathbb{R}^{T' \times d_z}$, where $T' = T/w$ denotes the frame numbers of the downsampled features. Then, we obtain the quantized sequence $\boldsymbol{Z_q}$ via an element-wise quantization function $Q$ that maps each element in $\boldsymbol{Z_q}$ to its closest codebook entry:

$$\boldsymbol{Z_q} = Q(\hat{\boldsymbol{Z}}) := (\arg\min_{\boldsymbol{z_k} \in \mathcal{Z}} \|\hat{\boldsymbol{z}}_t - \boldsymbol{z_k}\|_2) \in \mathbb{R}^{T' \times d_z}. \tag{2}$$

The quantized sequence is then reconstructed by the VQ-VAE decoder. Three 1D convolutional layers comprise the VQ-VAE decoder.

**Pose GPT.** With the learned discrete pose space, we build a Pose GPT network Vaswani et al. (2017); Siyao et al. (2022) that maps the speech and pose style embedding into the discrete codes of the target poses. The HuBERT feature and mel spectrogram of input speech are sent into several Transformer layers (Vaswani et al., 2017) as the speech content features. HuBERT feature conveys more linguistic information while mel spectrogram contains prosody information (Chen et al., 2022). We utilize these two features for the reason that head poses are correlated to both the speech semantics and the speech prosody (*e.g.,* stress, tone). We assign each pose sample in the training dataset with a one-hot label as the pose style embedding. It is passed through several linear layers, concatenated with the speech content features, and then fed into the Pose GPT comprising six cross-conditional Transformer layers. The predicted codes are finally sent into the VQ-VAE decoder to obtain the continuous head pose sequence. We train this Pose GPT using the teacher-forcing training scheme with the cross-entropy loss.

**Semantic-Aware Pose Style Matrix.** Given a speech utterance of $T$ frames and the corresponding head pose sequence, HuBERT (Hsu et al., 2021) assigns each speech frame into one of 512 clusters, while the VQ-VAE encoder transforms the pose sequence into the latent sequence $\hat{\boldsymbol{Z}}$. Thus, each frame in $\hat{\boldsymbol{Z}}$ is labeled with a cluster which contains the speech semantic information. We calculate the semantic-aware pose style matrix $S \in \mathbb{R}^{512 \times d_z}$ for each pose sequence as:

$$S_j = \sum_{i=1}^{T'} \hat{\boldsymbol{Z}}_i \cdot \delta(L_i, j) / \sum_{i=1}^{T'} \delta(L_i, j), j = [1, 2, \dots, 512], \tag{3}$$

where $L_i$ denotes the cluster label for $i$ frame of $\hat{\boldsymbol{Z}}$ and $\delta$ is the Kronecker symbol that outputs 1 when the inputs are equal else 0. This equation aggregates all the poses which have the same

semantic unit of HuBERT. Since the clustering step in HuBERT models the semantic information in acoustic units and the VQ-VAE encoder models the temporal pose style information, the derived $S$ represents the pose style that considers semantics.

**Procedure of the Pose Adapter.** The pretrain, adaptation, and inference procedures of the pose adapter are listed in Alg. 2. Note that the adaptation stage involves no parameter updating.

---

**Algorithm 2** Pretrain, Adaptation, and Inference of Pose Adapter.

---

1: **Pretrain:** Assign each pose sequence in the source training dataset with a one-hot pose style embedding. Train the VQ-VAE and Pose GPT with source training dataset. Calculate the semantic-aware pose style matrix for each sequence in the training dataset and establish the database for all pose style matrices and their corresponding pose style embedding.
2: **Adaptation:** Given an unseen pair of reference speech and pose sequence, calculate the semantic-aware pose style matrix for the reference data. Find the nearest pose style matrix in the established database by computing the L1 distance, and adopt the corresponding pose style embedding for inference.
3: **Inference:** Given a driving speech, and utilizing the retrieved pose style embedding, the pose adapter generates head poses with the pre-trained Pose GPT and the VQ-VAE decoder.

---

## 4 EXPERIMENTS

**Dataset.** Due to the scarcity of 3D animation datasets with personalized talking styles, we utilize the 2D audio-visual talking face dataset. For expression adapter pre-train, we utilize Obama weekly footage (Suwajanakorn et al., 2017) with a neutral talking style. For pose adapter pre-train, Vox-Celeb2 (Chung et al., 2018) is utilized, which contains about 1M talking-head videos of different celebrities with diverse and natural head poses. In the adaptation stage, we introduce MEAD (Wang et al., 2020) for the expression adapter, since MEAD consists of videos featuring 60 actors with 7 different emotions. We introduce VoxCeleb2 test dataset for the adaptation of the pose adapter.

**Dataset Processing.** We resample the 2D videos as 25fps and employ the 3D face reconstruction method DECA (Feng et al., 2021) to extract the identity, expression, and head pose. Given these three parameters, we can compute the 3D vertices of the per-frame meshes. For the audio track, we downsample the raw wave as 16000Hz and adopt the base HuBERT model (Hsu et al., 2021) to extract speech features. Since the frame rates of video and audio features are different, we interpolate the audio features and align them to the same length (Cudeiro et al., 2019).

**Baseline Methods.** We compare AdaMesh with several state-of-the-art methods. 1) FaceFormer (Fan et al., 2022), which first introduces autoregressive Transformer-based architecture for speech-driven 3D facial animation. 2) CodeTalker (Xing et al., 2023), which is the first attempt to model facial motion space with discrete primitives. 3) GeneFace (Ye et al., 2023), which proposes a domain adaptation pipeline to bridge the domain gap between the large corpus and the target person video. It can also generate realistic head poses with a template pose sequence. 4)Imitator (Thambiraja et al., 2023), which learns identity-specific details from a short input video and produces facial expressions matching the identity-specific speaking style. 5)FACIAL (Zhang et al., 2021), which focuses on predicting dynamic face attributes, including head poses. GeneFace and FACIAL are photo-realistic talking face methods, we therefore reconstruct 3D meshes Feng et al. (2021) from the generated videos for comparison.

## 5 RESULTS

### 5.1 QUANTITATIVE EVALUATION

**Evaluation Metric.** To measure lip synchronization, we calculate the lip vertex error (LVE), which is widely used in previous methods (Fan et al., 2022; Thambiraja et al., 2023). Since MEAD dataset contains significant emotion, we follow Peng et al. (2023) to calculate the emotional vertex error (EVE) to evaluate if the method can capture the personalized emotional talking style. The diversity scores (Siyao et al., 2022; Chen et al., 2023) for expressions (Div-$\delta$) and poses (Div-$\psi$) are separately computed to evaluate if the methods generate vivid and diverse motions. The landmark

Table 1: Quantitative evaluation and user study results. User study scores with 95% confidence intervals. Exp-Richness value reveals the percentages of users who prefer the compared method other than AdaMesh in A/B tests. Value lower than 50% indicates that users prefer AdaMesh.

| | Facial Expression | | | Head Pose | | | User Study | | |
|---|---|---|---|---|---|---|---|---|---|
| | LVE↓ | EVE↓ | Div-$\delta$↑ | FID↓ | LSD↑ | Div-$\psi$↑ | Lip-Sync | Pose-Natural | Exp-Richness |
| FaceFormer | 4.67 | 3.46 | 6.42 | 1.05 | 0.0 | 0.0 | 4.05±0.15 | 1.46±0.10 | 0.80% |
| Codetalker | 4.43 | 3.30 | 6.48 | 1.05 | 0.0 | 0.0 | 4.10±0.14 | 1.42±0.11 | 0.80% |
| FACIAL | 3.44 | 2.48 | 6.32 | 1.68 | 0.69 | 1.50 | 3.88±0.21 | 3.85±0.20 | 11.7% |
| GeneFace | 3.54 | 2.47 | 6.56 | 1.38 | 0.70 | 1.49 | 4.01±0.18 | 3.79±0.17 | 12.3% |
| Imitator | 3.15 | 2.33 | 5.97 | 1.05 | 0.0 | 0.0 | 3.82±0.18 | 1.42±0.11 | 29.8% |
| Ground Truth | – | – | 7.85 | – | 1.37 | 3.01 | 4.30±0.14 | 4.12±0.23 | 41.3% |
| AdaMesh (Ours) | **2.91** | **2.25** | **7.10** | **0.90** | **0.87** | **1.57** | **4.12±0.19** | **4.22±0.11** | – |

standard deviation (LSD) (Zhang et al., 2021; Xing et al., 2023) is used to measure the pose dynamics. The FID score (Siyao et al., 2022; Zhu et al., 2023) for motion sequence is also employed to estimate the pose realism.

**Evaluation Results.** The results are shown in Tab.1. For facial expressions, it can be observed that our method obtains lower LVE than other methods, which indicates AdaMesh has more accurate lip movements. FaceFormer and CodeTalker have poorer performance, since the LVE score is calculated on MEAD and Obama datasets, and these two methods have poor generalizability to unseen speaker identity. AdaMesh also achieves the lowest EVE. It suggests that our approach can efficiently capture the facial style in the adaptation. A higher diversity score for expressions confirms that our approach generates more vivid facial dynamics. For head poses, FaceFormer, CodeTalker, and Imitator cannot generate head poses. GeneFace incorporates a real head pose sequence for presentation and FACIAL learns over-smoothed poses. The FID scores show that our approach generates more realistic head poses, while LSD and diversity scores confirm our approach generates head poses with higher diversity.

## 5.2 QUALITATIVE RESULTS

Since quantitative metrics cannot fully reflect the perceptual observation, we conduct qualitative evaluation from three aspects, as shown in Fig. 3.

**Speech-Lip Synchronization.** To evaluate the lip synchronization performance, we illustrate three keyframes speaking the vowels and consonants. The visual results show that our approach generates more accurate lip movements than other methods. AdaMesh has more pronounced mouth open when speaking /ou/ and /ei/. FaceFormer and CodeTalker close their mouth tighter on /b/ than other methods, even the ground truth, since they are directly trained on the 3D scanned vertices. Other methods are trained or fine-tuned on the reconstructed data from 2D video, which are less precise than the scanned data.

**Facial Expression Richness.** To evaluate if the methods can capture the facial talking style from the reference video. We compare GeneFace, Imitator, and our AdaMesh on MEAD dataset. It can be observed that our approach preserves the expressive facial style and obviously gains richer facial details, especially around the cheek and lip regions. AdaMesh even gains more expressive details and accurate lip movements than the ground truth. The MoLoRA parameters in AdaMesh act as a plugin of facial style for the pre-trained model, and the pre-trained model on a larger scale data has guaranteed lip synchronization.

**Head Pose Naturalness.** We adapt our pose adapter on the VoxCeleb2-Test dataset. As shown in Fig. 3(c), we illustrate frames in a sentence that contains evident emotion. A sentence that conveys strong emotions is accompanied by more evident head poses. Our approach produces head movements with a greater range of amplitudes, and the results are closer to the ground truth. Following FACIAL (Zhang et al., 2021), we also plot the landmark tracemaps for a sentence that conveys negative semantics. The head poses with more degrees of freedom confirm the diversity of AdaMesh.

**User Study.** We conduct a user study to measure the perceptual quality of these methods. Fifteen users with professional English ability participate in this study. For lip synchronization and pose

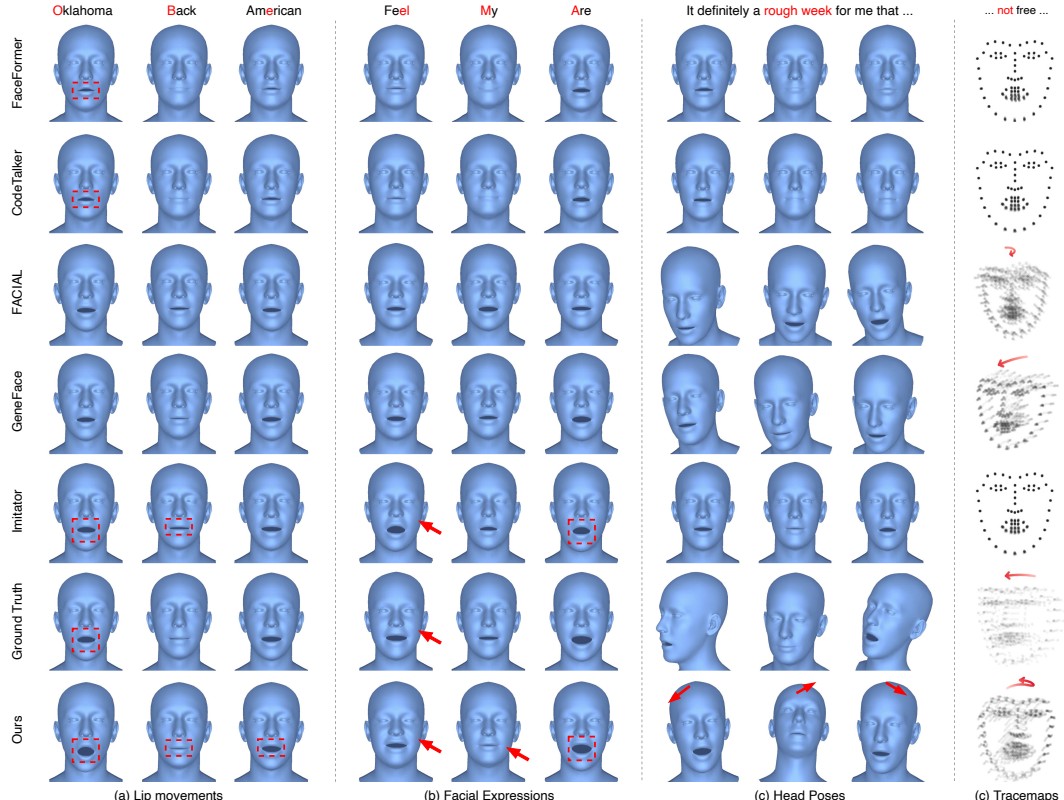

Figure 3: Qualitative comparison with different methods. (a) shows lip movements on Obama dataset with neutral talking style. (b) is for observation of facial expressions on the emotional MEAD dataset. (c) and (d) show head poses and corresponding landmark tracemaps on the VoxCeleb2-Test dataset. The first row displays the words or sentences that these frames are pronouncing.

naturalness, we adopt mean-opinion-score (MOS) tests to ask users to score each video from one (worst) to five (best). Our approach has achieved the highest score on these two aspects. For expression richness, we adopt A/B test to ask users to select a better video between AdaMesh and another method. Our approach has significantly outperformed other methods, including the ground truth. This is attributed to the image jittering brought by Feng et al. (2021); Li et al. (2017). AdaMesh has filtered out these disturbances in the pre-training of the expression adapter.

## 5.3 APPROACH ANALYSIS

**Ablation Study for Expression Adapter.** For expression adapter, we separately remove the style encoder, identity encoder, and MoLoRA to verify the necessity of these modules. 1) As shown in Tab. 2, the removal of the style encoder leads to a drop of vertices errors and diversity metrics by a large margin. It suggests that the style encoder provides essential details which cannot be predicted from speech. 2) The absence of identity also causes a slight performance drop since the expression and identity parameters are not fully disentangled (Li et al., 2017; Feng et al., 2021). The identity parameters also contain person-specific talking style. 3) We fine-tune the full expression adapter instead of updating MoLoRA parameters. It can be observed that the performance significantly decreases since fine-tuning with few data damages the generalization ability of the pre-trained model.

**Ablation Study for Pose Adapter.** For pose adapter, we test three settings. 1) The retrieval of the pose style matrix is removed by randomly selecting a pose style embedding in the training data for inference. All metrics drop significantly due to the unawareness of the reference's personalized talking style. 2) w/o HuBERT feature, where only the mel spectrogram is utilized for the speech representation. The LSD and diversity scores show that HuBERT features contain semantic information, which is vital to the diverse head pose generation. 3) After removing VQ-VAE, the model

Table 2: Ablation for expression adapter.

| Setting | LVE↓ | EVE↓ | Div-$\delta$↑ |
|---|---|---|---|
| Ours | **2.91** | **2.25** | **7.10** |
| w/o style encoder | 3.12 | 2.34 | 6.33 |
| w/o identity encoder | 3.05 | 2.29 | 6.98 |
| w/o MoLoRA | 3.10 | 2.39 | 6.21 |

Table 3: Ablation for pose adapter.

| Setting | FID↓ | LSD↑ | Div-$\psi$↑ |
|---|---|---|---|
| Ours | **0.90** | **0.87** | **1.57** |
| w/o style matrix | 1.01 | 0.69 | 0.69 |
| w/o HuBERT feature | 0.91 | 0.07 | 0.15 |
| w/o VQ-VAE | 0.91 | 0.05 | 0.02 |

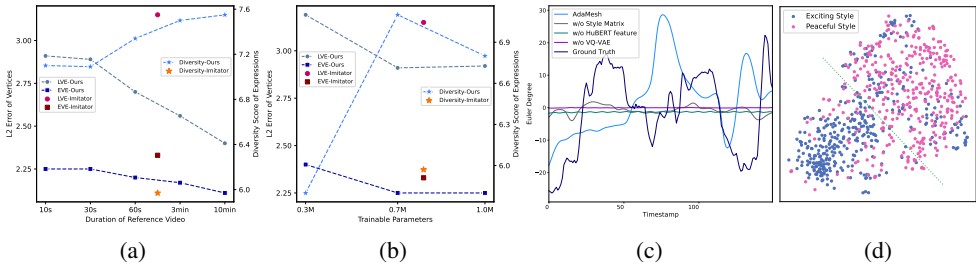

(a)      (b)      (c)      (d)

Figure 4: Approach analysis. (a) performance vs. amount of adaptation data. (b) performance vs. amount of trainable parameters. (c) yaw degree curves for head poses. (d) t-SNE visualization of the semantic-aware pose style matrices.

directly predicts the continuous pose values. The LSD and diversity scores drop to near zero. It suggests that VQ-VAE provides realistic quantized codes which is easier to predict. To further investigate the efficacy, we plot the yaw degree of head poses in Fig. 4c. The pose curve of AdaMesh has the largest variation, which is closer to the ground truth. It confirms the results in quantitative and qualitative experiments. We also utilize t-SNE (Van der Maaten & Hinton, 2008) to visualize the pose style matrix for the selected exciting and peaceful head poses. The visualization result shows that different styles are clustered into different groups. This proves that the semantic-aware pose style matrix contains meaningful pose style information.

**Amount of Adaptation Data.** We only analyze the impact of the amounts of adaptation data on the performance of the expression adapter, since the pose adapter actually is a zero-shot model with the retrieval strategy. Fig. 4a demonstrates the performance vs. adaptation data amount for AdaMesh. Points within the same color family represent the same method, while the same marker corresponds to the computation of the same metric. Thus the points with the same marker can be compared. There are two y-axis in the figure, and the legend next to each y-axis indicates what metric this y-axis represents. We can conclude from this figure that, increasing the amount of adaptation data can obviously improve the performance. AdaMesh achieves significantly better performance than Imitaor with the same amount of adaptation data.

**Amount of adaption Parameters.** To better estimate the impact of the amounts of MoLoRA parameters on the performance of the expression adapter, we evaluate the performance of MoLoRA with different rank size combinations and the vanilla LoRA (the point in Fig. 4b with the maximum number of parameters). The point with the highest diversity score corresponds to the rank sizes of $[4, 8, 16, 32]$. This setting also achieves the lowest LVE and EVE scores. Furthermore, when optimizing roughly the same number of parameters, our approach outperforms Imitator in all metrics.

## 6   CONCLUSION

This paper proposes AdaMesh for speech-driven 3D facial animation, which aims to capture the personalized talking style from the given reference video, to generate rich facial expressions and diverse head poses. Our central insight is to separately devise suitable adaptation strategies for facial expressions and head poses. The MoLoRA and retrieval adaptation strategies are in line with the characteristics of these two types of data. The proposed approach successfully solves the catastrophic forgetting and overfit problems for facial expression modeling and the averaged generation problem for head pose modeling under the circumstance of few data adaptation. Extensive experimental results and approach analysis prove that AdaMesh archives high-quality and efficient style adaptation, and outperforms other state-of-the-art methods.

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

# A APPENDIX

This supplemental material contains three subsections: the details of the expression adapter and the pose adapter in AdaMesh, implementations of baseline methods, and additional information about the user study.

## A.1 DETAILS OF EXPRESSION ADAPTER AND POSE ADAPTER

The architecture details of AdaMesh are presented in Tab. 4. The overall architecture of the expression adapter is illustrated in Fig. 1 of the main paper. For the identity encoder, $T$ frames identity parameters are fed into two convolutional layers and average to a vector with a dimension of 256. The speech sequence and expression sequence are separately processed by the audio encoder and style encoder. These two encoders are three conformer layers to output features with $T \times 256$ dimension. The encoded features are concatenated with the identity latent feature and sent into the expression decoder. We also adopt three Conformer layers for the decoder to output the predicted expression sequence. This is a simple architecture, but with robust and efficient performance, which is mainly attributed to the identity encoder and the style encoder providing the residual style information.

For the pose adapter, following Siyao et al. (2022), VQ-VAE downsample and upsample the pose sequence with a symmetrical network structure of 1D convolutional layers. We also adopt the cross-conditional causal attention mechanism other than the vanilla attention in Transformer, since the head poses of different degrees are entangled with each other. We present the loss function of VQ-VAE as below:

$$\mathcal{L}_{VQ} = \mathcal{L}_{rec}(\hat{\boldsymbol{\psi}}, \boldsymbol{\psi}) + \|sg[\hat{z}] - z_q\| + \beta \|z - sg[z_q]\| \tag{4}$$

where $\hat{\psi}$ and $\psi$ denotes output euler angle sequence and input ground truth, respectively. And the specific formula for $\mathcal{L}_{\text{rec}}$ can be expressed as:

$$\mathcal{L}_{\text{rec}}\left(\hat{\psi}, \psi\right) = \left\|\hat{\psi} - \psi\right\|_1 + \alpha_1 \left\|\hat{\psi}' - \psi'\right\|_1 + \alpha_2 \left\|\hat{\psi}'' - \psi''\right\|_1 \tag{5}$$

where $\psi'$ and $\psi''$ represent the 1st-order (velocity) and 2nd-order (acceleration) partial derivatives of euler angle sequence $\psi$ on time. In addition, $\alpha_1$, $\alpha_2$ and $\beta$ denote trade-off coefficients while $sg[\cdot]$ denotes "stop gradient".

Table 4: Architecture parameter list for AdaMesh.

| | | | |
|---|---|---|---|
| Expression Adapter | Identity Encoder | Conv1d Kernel | 5 |
| | | Conv1d Channel Size | 256 |
| | | Identity Embedding Width | 256 |
| | Audio Encoder | Conformer Layer | 3 |
| | | Conformer Layer Head | 4 |
| | | Conformer Layer Embedding Width | 256 |
| | Style Encoder | Audio Embedding Width | 256 |
| | | Conformer Layer | 3 |
| | | Conformer Layer Head | 4 |
| | | Conformer Layer Embedding Width | 256 |
| | | Audio Embedding Width | 256 |
| | Expression Decoder | Conformer Layer | 3 |
| | | Conformer Layer Head | 4 |
| | | Conformer Layer Embedding Width | 256 |
| | | Audio Embedding Width | 256 |
| | | Conv1d Kernel | 5 |
| | | Conv1d Channel Size | 256 |
| Pose Adaptor | VQ-VAE | Encoder Layers | 3 |
| | | Decoder Layers | 3 |
| | | Encoder/Decoder Conv1d Kernel | 4 |
| | | Encoder/Decoder Conv1d Channel Size | 256 |
| | | Codebook Size | 1024 |
| | | Codebook Embedding Width | 256 |
| | Pose GPT | Transformer Layers | 6 |
| | | Transformer Layer Head | 12 |
| | | Transformer Layer Embedding Width | 768 |
| | | Style Embedding Width | 50 |

## A.2 IMPLEMENTATIONS OF BASELINE METHODS

As mentioned in the main paper, we compare AdaMesh with state-of-the-art methods. FaceFormer (Fan et al., 2022) and CodeTalker (Xing et al., 2023) have released the pre-trained model, thus we test inference with the available codes. To calculate metrics with the same identity topology, we adopt Li et al. (2017) to recover the shape parameter from 2D video are convert it to the format that FaceFormer and CodeTalker require for inference. For GeneFace (Ye et al., 2023), we extract the required data to train the image render and fine-tune the post-net. For FACIAL (Zhang et al., 2021), since it is a person-specific method which requires a large amount of data for training, we directly utilize the pre-trained checkpoint of Obama for inference. For Imitator (Thambiraja et al., 2023), it is the most-related and latest work. We implement it to the best of our understanding. As soon as Imitator releases the public codes, we will replace the visualized images in our paper with the original ones.

## A.3 USER STUDY

The designed user study of the mean-of-score tests for lip synchronization and head pose naturalness and the A/B tests for the richness of facial expressions are separately demonstrated in Fig. 5 and Fig.

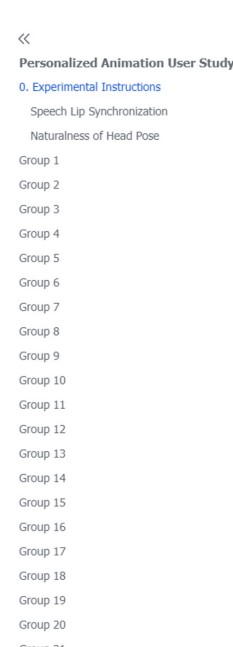

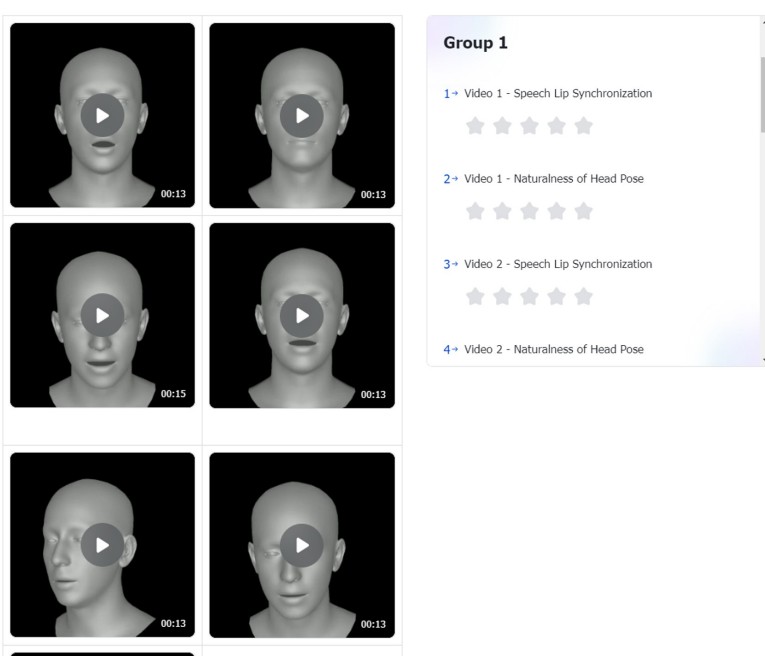

Figure 5: Designed mean-of-score test interface for lip synchronization and head pose naturalness.

6. Prior to commencing the tests, we have thoroughly communicated the experimental guidelines and important information that users need to be aware of. For the lip synchronization and head pose naturalness tests, we sample 105 videos for seven compared methods. For the A/B tests, 60 pairs of compared videos are created.

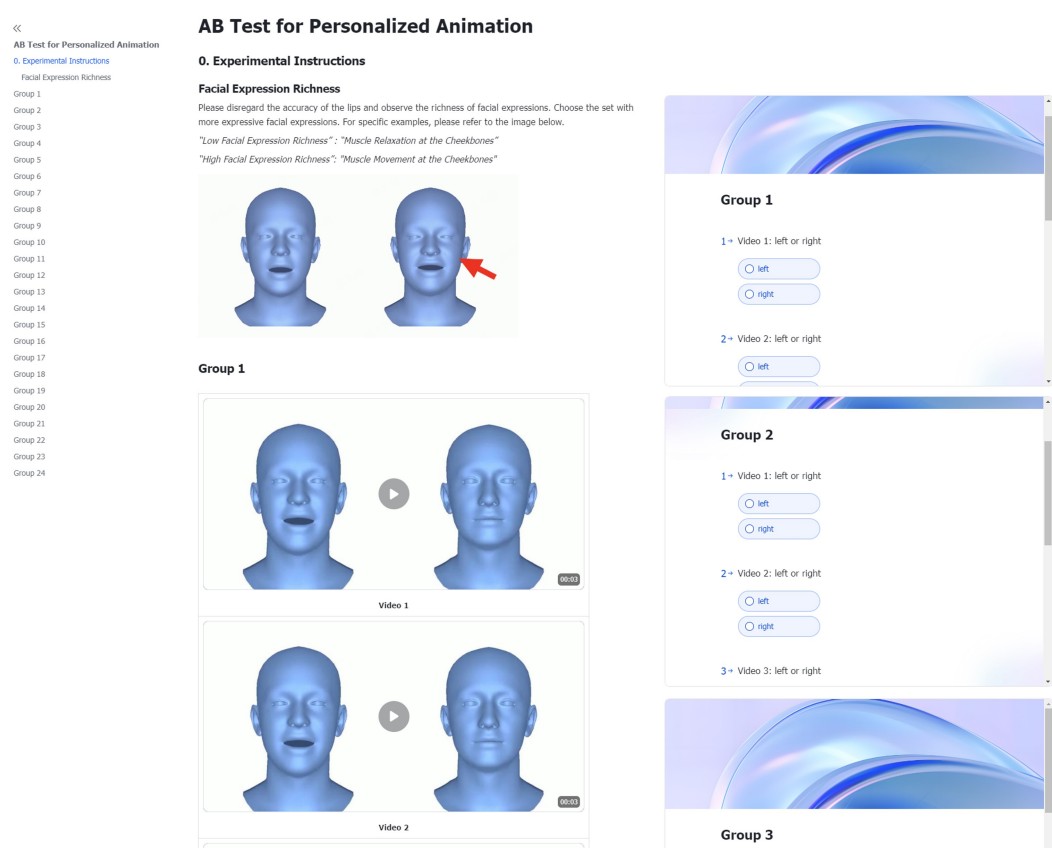

Figure 6: Designed A/B test interface for the richness of facial expressions.

