# OpenReview forum: "Personalized Facial Expressions and Head Poses for Speech-Driven 3D Facial Animation"
_ICLR.cc/2024/Conference — ICLR 2024 Conference Withdrawn Submission_

### Official Review · Reviewer_ChZu · 2023-10-29

**Soundness:** 3 good
**Presentation:** 3 good
**Contribution:** 3 good
**Rating:** 8
**Confidence:** 4

**Summary:**

The paper proposes a novel architecture for speech-driven 3D facial animation amenable to pose and expression inputs. A series of encoders are properly combined to ensure the generation abides to the desired pose and facial expressions. The paper includes some novel components in the domain of face synthesis, namely the use of LoRA for convolutions to facilitate the training, and the use of a codebook with a PoseGPT to generate a plausible sequence of poses. The authors include a very strong supplementary material showcasing the benefits of the proposed approach, and a user-study to support the generated results.

**Strengths:**

The paper is overall well written and presented, and it is accompanied by a thorough supplementary material with a detailed video showcasing the results. The amount of work put towards completing the paper is significant and valuable.

- The attempt to encode facial expressions and pose while synthesizing 3D faces from speech is novel as far as I am aware, and the results are generally remarkable.

- The use of a VQ-VAE to quantize poses and a generative method to generate these from the initial latent vector is also novel and (seems to be) technically sound. This generative approach from a codebook is also well-grounded and allows for generating meaningful poses across sequences.

- The use of LoRA to facilitate the learning is also novel in this context, and while I am not sure such plug and play can be counted as a contribution, its adaptation to convolutions is interesting.

**Weaknesses:**

In my opinion, there are only few aspects that could need further consideration:

1.) It is my understanding that the facial expressions are passed as input to the network, is that correct? If this is the case, I would like the authors to provide some visual examples of the same inputs where the target facial expressions are different. I wonder how this combines to the fact that speech is also a cue conveying emotions and thus cannot be 100% detached from facial expressions. If the input video is of a person in a state of sadness, how would the method perform when the reference expression is set to happy? How realistic would this be? If the expression is not directly inferred from the audio, then I think Fig. 1 needs a better representation and this to be properly referred to in the text.

2.) The method contains an identity encoder. I wonder how is this used when generating the outputs considering that the final result is a 3D mesh without texture. In the case that this affects the 3D mesh only, I think that a study showing whether humans can distinguish two different generated meshes of the same person from a third one is necessary. Generating 3D meshes without regards to proper identity preservation can lead to relatively poor user experience.

**Questions:**

I have included all my questions above in the weaknesses section. Please do address these.

---

> ### Author Response · Authors · 2023-11-13
> **Author Response to Reviewer ChZu**
>
> We thank the reviewer for the constructive feedback and considering our work as technically sound. We hope our response resolves your concerns fully.
>
> > **Q1**:  Is it correct that the facial expressions are passed as input to the network? If this is the case, I would like the authors to provide some visual examples of the same inputs where the target facial expressions are different. I wonder how this combines to the fact that speech is also a cue conveying emotions and thus cannot be 100% detached from facial expressions. If the input video is of a person in a state of sadness, how would the method perform when the reference expression is set to happy? How realistic would this be? If the expression is not directly inferred from the audio, then I think Fig. 1 needs a better representation and this to be properly referred to in the text.
>
> **A1**: As understood by the reviewer, the facial expressions are passed into the network to provide style information. The same speech input can produce different facial expressions, as can be observed from the demo video at 3:02. In the proposed expression adapter, the driving speech provides content information while the reference expressions provide style information. Thus, if the driving audio is sad and the reference expression is happy, the generation will be happy. It's a good suggestion that we emphasize the text/content information provided by driving speech in Fig.1. We will polish Fig. 1 in the revised paper.
>
> > **Q2**: The method contains an identity encoder. I wonder how is this used when generating the outputs considering that the final result is a 3D mesh without texture. In the case that this affects the 3D mesh only, I think that a study showing whether humans can distinguish two different generated meshes of the same person from a third one is necessary. Generating 3D meshes without regards to proper identity preservation can lead to relatively poor user experience.
>
> **A2**: The input of the identity encoder is the identity parameters extracted by the 3D face model [1, 2] from the RGB video. The identity parameters typically refer to the unique geometric features of a face, essentially the fundamental structure that constitutes an individual's facial characteristics. These parameters capture the variations in facial shape among individuals, such as the length and width of the face, the shape of the nose, and the size and positioning of the eyes.  The reason for introducing an identity encoder is that, the person's identity contains personality, which cannot be fully disentangled from the expression parameters.  It's true as the reviewer recommends that leaving identity will lead to poor user experience, and it needs more investigation.
>
> [1] Tianye Li, et al. Learning a Model of Facial Shape and Expression from 4D Scans. ACM Trans. Graph., 36(6), 2017.
> [2] Yao Feng, et al. Learning an Animatable Detailed 3D Face Model from In-The-Wild Images. ACM Trans. Graph., 40(4), 2021.

---

### Official Review · Reviewer_aeAB · 2023-10-30

**Soundness:** 3 good
**Presentation:** 2 fair
**Contribution:** 2 fair
**Rating:** 3
**Confidence:** 5

**Summary:**

This paper proposes novel adaption strategy for capturing the reference emotion and pose styles in 3D talking head generation. Given a reference video and a driving speech, the proposed AdaMesh method can generate a 3D talking head saying the speech with dynamic emotions and poses similar to the reference video. The expression and pose are generated by separate network branches, which features mixture-of-LoRA (MoLoRA) adaption and discrete pose space, respectively.

**Strengths:**

### 1. Novel MoLoRA for personalized facial expression

LoRA may currently be the most appropriate technic for adapt a generalized model into personalized (or case-specific) ones. The verification of LoRA in the talking head field should benefit the community.

### 2.  Investigation of semantic-aware pose style

Pose functions as an information conveying modal in human communication, while the relation of speech semantic and head pose lacks investigation in the talking head/face researches. This paper propose a novel strategy to align semantic features into pose codes, in which the pose codebook can be established using large pose datasets without speech.

### 3. Better demo results than other compared works

The qualitative results are remarkably better than others in the demo video, including more realistic poses and prominent expressions.

**Weaknesses:**

1. I argue with the authors' claim that "**the first work that takes both facial expressions and head poses into account in personalized speech-driven 3D facial animation.**" Facial[r1] meets all the key words, **facial expression**, **head pose**, **personalized**, and **3D**. Besides, SadTalker[r2] implements specific branches for pose and expression generation for 3DMMs, which has not been cited. The SadTalker finally render 2D RGB videos from its generated 3DMMs, which I think cannot cover up that it's a 3D method and exclude it from the discussion in the reviewing paper.
2. Although the generated poses are better than compared works, they are **not realistic enough to convey semantics**, thus I cannot distinguish whether the poses are semantic-aware as claimed in paper. The generated poses drive the shoulder with the same displacement and rotation with the head, while human rotate their heads with relatively fixed shoulders. The dynamics of neck needs more investigation.
3. The pose adaption(retrieval) method means it **can only generate seen pose styles** in the training set.
4. The **sampling method** of Pose-GPT is missing.
5. Minor issue: in the last line of Sec. 4 Dataset, "We **introduce** VoxCeleb2 test dataset ..." I think "introduce" is inappropriate since the VoxCeleb2 is already mentioned for pose adapter pre-train.

[r1] Zhang, Chenxu, et al. "Facial: Synthesizing dynamic talking face with implicit attribute learning." Proceedings of the IEEE/CVF international conference on computer vision. 2021.
[r2] Zhang, Wenxuan, et al. "SadTalker: Learning Realistic 3D Motion Coefficients for Stylized Audio-Driven Single Image Talking Face Animation." Proceedings of the IEEE/CVF Conference on Computer Vision and Pattern Recognition. 2023.

**Questions:**

Do you assign a number to each semantic-aware pose style matrix S and transform this very number into a one-hot vector as pose style embeddings?

---

> ### Author Response · Authors · 2023-11-14
> **Author Response to Reviewer aeAB**
>
> Dear reviewer, we sincerely appreciate your careful understanding of our approach, and thanks for your meaningful opinion. We hope our response fully resolves your concerns.
>
> > **Q1**: I argue with the authors' claim that "the first work that takes both facial expressions and head poses into account in personalized speech-driven 3D facial animation." FACIAL and SadTalker cannot be excluded from the discussion.
>
> **A1**: We made this announcement following the Section 5 limitation discussion in EmoTalk [1], which is the latest work in speech-driven 3D animation. From the perspective of the entire talking face generation field, we agree with the reviewer that SadTalker should also be included as modeling both facial expressions and head poses. **We conduct new comparison with SadTalker, and still gain significant superiority** over SadTalker on facial expressions and head poses. We will add the new results to the revised paper.
>
> > **Q2**: Although the generated poses are better than compared works, they are not realistic enough to convey semantics, thus I cannot distinguish whether the poses are semantic-aware as claimed in paper. The generated poses drive the shoulder with the same displacement and rotation with the head, while human rotate their heads with relatively fixed shoulders. The dynamics of neck needs more investigation.
>
> **A2**: **The semantics for head poses can be observed from the demo video.** For example, when saying the word **"rough" at 2:12** and **"not" at 2:21**, there was **a noticeable turn of the head**; when the **voice pitch rises at 2:25**, the head noticeably **tilts upwards**. For the neck rotation, it is a great suggestion by the reviewer that shoulder should be fixed. However, limited by the adopted face model [2], we cannot separately control the head and neck. We will try other face models for the neck modeling in the future work, and we believe that the strategy of the pose adapter can be applied to the neck modeling.
>
> > **Q3**: The pose adaption (retrieval) method means it can only generate seen pose styles in the training set.
>
> **A3**: The retrieval strategy for pose adaption does mean that it can only generate seen pose styles in the training set. However, the **VQ-VAE is trained on a large-scale dataset, which contains more than 100 million videos of 6000 speakers.** We believe that **most pose styles are covered in this dataset.** Additionally, the head poses are mostly associated with the conveyance of negation, affirmation, and turnaround in semantics. The retrieval strategy is a better choice than fine-tuning.
>
> > **Q4**: The sampling method of Pose-GPT is missing.
>
> **A4**: In the inference process of PoseGPT, it begins with an initial start symbol and autoregressively outputs discrete tokens of the pose with given the speech representation. The principle for generating each token is to **select the output with the highest probability at each step.** We will add it to the revised paper.
>
> > **Q5**: Minor issue: in the last line of Sec. 4 Dataset, "We introduce VoxCeleb2 test dataset ..." I think "introduce" is inappropriate since the VoxCeleb2 is already mentioned for pose adapter pre-train.
>
> **A5**: Thanks for pointing this out. We will correct it in the revised version.
>
> > **Q6**: Do you assign a number to each semantic-aware pose style matrix S and transform this very number into a one-hot vector as pose style embeddings?
>
> **A6**: Yes, as the reviewer understands, we assign each pose sequence in the training dataset with a one-hot pose style embedding, and calculate the semantic-aware pose style matrix for each pose sequence. We establish the database for all pose style matrices and their corresponding pose style embedding for efficient retrieval.  This procedure can be found in Algorithm 2. **We choose to condition the pose generation on the one-hot embedding instead of the pose style matrix, since the different derivations of the pose style matrix will not influence the pose generation.** The pose style matrix is only used for pose style adaptation.
>
> [1] Ziqiao Peng, et al. EmoTalk: Speech-Driven Emotional Disentanglement for 3D Face Animation. In Proceedings of the IEEE/CVF International Conference on Computer Vision. 2023.
> [2] Tianye Li, et al. Learning a Model of Facial Shape and Expression from 4D Scans. ACM Trans. Graph., 36(6), 2017.

---

### Official Review · Reviewer_UjL5 · 2023-10-31

**Soundness:** 2 fair
**Presentation:** 3 good
**Contribution:** 2 fair
**Rating:** 5
**Confidence:** 5

**Summary:**

This research introduces AdaMesh, a approach to speech-driven 3D facial animation that learns personalized talking styles from short reference videos, resulting inexpressive facial expressions and head poses. The proposed method, which includes MoLoRA for facial expression adaptation and a pose adapter with a semantic-aware pose style matrix, outperforms existing techniques.

**Strengths:**

1.AdaMesh addresses the limitation of existing works by focusing on capturing and adapting the individual's talking style, including facial expressions and head pose styles.

2.AdaMesh introduces a technique called mixture-of-low-rank adaptation (MoLoRA) for fine-tuning the expression adapter.

3.A pose GPT and VQ-VAE are used for the pose adapter.

**Weaknesses:**

1.The major contributions are MoLoRA for facial expression adaptation and pose GPT for pose adaptation. The two key points have no much correlation between each other, the combination of these two points makes this paper not well-focused.


2.Given the head pose training data from VoxCeleb2, how do you extract the head pose information? The GT head pose in the demo video seems unstable and lacks time consistency. In such a condition, does the model learn to generate meaningful head pose?

3.Does the authors try to train baselines, such as FaceFormer or CodeTalker, on their data?

4.Experiments on the classical datasets, such as the VOCA dataset, could further verify the model’s effectiveness. The results on one dataset can not prove the model’s generalization.

5.User study lacks credibility, more details about the user study setting should be clarified.

**Questions:**

see above

---

> ### Author Response · Authors · 2023-11-13
> **Author Response to Reviewer UjL5**
>
> Dear reviewer, we sincerely appreciate your careful understanding of our approach, and thanks for your meaningful opinion. We hope our response fully resolves your concerns.
>
> > **Q1**: The combination of the two major contributions makes this paper not well-focused.
>
> **A1**: **The two major contributions serve a common goal**, which focus on generating personalized facial expressions and head poses with limited adaptation data. **The central insight is to separately devise suitable adaptation strategies** for facial expressions and head poses. The adaptation strategy for expression should capture speech-lip correlation and talking style, thus fine-tuning is an appropriate choice. For the adaptation strategy for head pose, considering the strong and weak semantic associations in head pose patterns, retrieval is a better adaptation choice. **The combination of these two designs contributes to significantly better performance.**
>
> > **Q2**: How do you extract the head pose information from VoxCeleb2. The GT head pose in the demo video seems unstable and lacks time consistency. In such a condition, does the model learn to generate meaningful head pose?
>
> **A2**: We utilize 3DDFA_V2 [1] for robust head pose extraction. The extraction is conducted at frame level, leading to the lack of time consistency. **The quantization of VQ-VAE is introduced to remove the instability between frames.** As you can observe from the demo video, the predicted head poses are more consistent and diverse than the ground truth.
>
> > **Q3**: Does the authors try to train baselines, such as FaceFormer or CodeTalker, on their data?
>
> **A3**: Yes, we retrain most baselines for evaluation, and we **select the better result for presentation between the retrained checkpoint and the original checkpoint**. We will give more comparison details. The retrained checkpoints for the 3D animation methods (i.e., FaceFormer, CodeTalker) achieve worse results than the original checkpoints since the limited training data compromises the model's generalizability. For GeneFace and Imitator, we retrain the posnet and motion decoder mentioned in the original papers. For FACIAL, we directly adopt the original checkpoint for inference, since FACIAL is a person-specific method. These details will be described in the supplementary materials in the revised paper.
>
> > **Q4**: Experiments on the classical datasets, such as the VOCA dataset, could further verify the model’s effectiveness. The results on one dataset can not prove the model’s generalization.
>
> **A4**: Thanks for the reviewer's constructive advice. We conduct additional experiments on the VOCA dataset and calculate the lip vertex error (LVE) for comparison. It can be seen that our approach achieves competitive performance, **although our approach is not trained on VOCA dataset**. This confirms the generalization ability of our approach.
> | Method | LVE (mm) | Trained on VOCASET |
> | :--- | :---: | :---: |
> | FaceFormer | 4.3011 | Yes |
> | CodeTalker | 4.2902 | Yes |
> | AdaMesh (Ours) | 4.2933 | **No** |
>
> > **Q5**: User study lacks credibility, more details about the user study setting should be clarified.
>
> **A5**: The main user study details are presented in Section 5.2 and supplementary material A.3. **We now describe more details and will add them to the revised paper.**
> We conduct a user study to measure the perceptual quality of these methods. Fifteen users with professional English ability participate in this study. For lip synchronization and pose naturalness, we adopt mean-opinion-score (MOS) tests to ask users to score each video from one (worst) to five (best). For expression richness, we adopt the A/B test to ask users to select a better video between AdaMesh and another method.
> - Lip Synchronization: How much do the lip movements match the audio? Very good (5) for no wrong lip movements and have nothing different from the ground-truth person talking. Worst (1) for the lip movements are totally unreasonable and cannot read content from the lips at all.
> - Pose Naturalness:  How real and natural are the head poses? Very good (5) for natural head shaking that aligns with semantics, such as noticeable trembling on emphasized words. Worst (1) for unacceptable performance where the head is always in a stationary or severely trembling state.
> - Expression Richness: Select the video which has richer facial expressions, with more pronounced cheek muscle movements, larger amplitude of mouth movements or observable facial emotion.
>
> [1] Jianzhu Guo, et al. Towards Fast, Accurate and Stable 3D Dense Face Alignment. In Proceedings of the European Conference on Computer Vision. 2020.

---

### Official Review · Reviewer_GUue · 2023-11-06

**Soundness:** 3 good
**Presentation:** 2 fair
**Contribution:** 3 good
**Rating:** 5
**Confidence:** 3

**Summary:**

This paper focuses on generating the talking face with a person-specific style, i.e., facial expressions and head poses. In the proposed method AdaMesh, the facial expression style is adapted by a mixture-of-low-rank adaptation. The pose style is adapted by matching the poses to discrete pose priors. Extensive experiments show that the proposed method preserves the talkers’ talking styles.

**Strengths:**

1. The proposed method preserves personalized talking styles, including facial expressions and head poses.
2. Qualitative and quantitative results show the advantages of the proposed AdaMesh.

**Weaknesses:**

1. This paper uses LORA with multiple ranks to capture the multi-scaled features of facial expressions. It is unclear why the multi-rank structure could build the multi-scale representation. Please explain their correlations.

2. This paper uses discrete poses as priors, and then during the inference this paper matches each pose into its nearest prior pose. It is unclear how to determine the number of prior poses. Intuitively, if the prior poses are sparse, it might influence the accuracy and smoothness of the generated poses. If the prior poses are dense, it requires plenty of training data to learn the generator for each pose prior.

3. The introduction of PoseGPT is not clear. What is the task when PoseGPT is trained? Fig 2 shows that poseGPT is trained by predicting the poses (pitch, yaw, roll) according to the driving speech and pose style embedding. Are the input pose style embeddings with the same length as that of the predicted head pose sequences? If not, how should we determine the length of the predicted head pose sequence?

4. Some minors:
  - The first sentence in the paragraph about Pose GPT lacks a  comma or period.

**Questions:**

Please refer to the weakness.

---

> ### Author Response · Authors · 2023-11-13
> **Author Response to Reviewer GUue**
>
> Dear reviewer, we sincerely appreciate your careful and accurate understanding of our approach. We hope our response fully resolves your concerns.
>
> > **Q1**: Why the multi-rank structure could build the multi-scale representation? Please explain their correlations.
>
> **A1**: The basic intuition is that the LoRA with small rank size filters out facial details and tends to preserve global features (e.g., emotion), while the LoRA with bigger rank size captures more fine-grained dynamics (e.g., muscle wrinkles). A similar intuition is confirmed of effectiveness for speech synthesis [1]. In the experiments, we empirically find that combining LoRA with different rank sizes benefits the model by generating facial expression of the finest details. It can be observed from Fig.4(b), the multi-rank LoRA achieves significant improvement on all metrics compared with other fine-tuning settings.
>
> > **Q2**: Unclear how to determine the number of prior poses.
>
> **A2**: The number of the discrete pose prior is 1024, since the reconstruction and perplexity loss for VQ-VAE under this setting are the lowest. This is a common way to select hyperparameters for VQ-VAE [2,3].  For pose generation, the discrete prior has removed instability in the extracted pose sequence and quantized poses into meaningful units. This design contributes to a more stable and diverse pose generation, and it actually has no influence on accuracy and smoothness.
>
> > **Q3**: The introduction of PoseGPT is not clear.
>
> **A3**: The pose adapter is composed of a VQ-VAE and a PoseGPT. The VQ-VAE encoder quantizes the ground-truth pose (pitch, yaw, roll) into discrete codes, and the VQ-VAE decoder reconstructs poses from the discrete codes. The PoseGPT is trained to predict discrete codes from the driving speech and the pose style embedding (a vector), and then the VQ-VAE decoder finally outputs the poses. The length of the predicted pose sequence is the same as the driving speech.
>
> > **Q4**: The first sentence in the paragraph about Pose GPT lacks a comma or period.
>
> **A4**: Thanks for pointing this out. We will correct it in the revised version.
>
> [1] Kaizhi Qian, et al. Unsupervised Speech Decomposition Via Tripple Information Bottleneck. In Proceedings of the International Conference on Machine Learning. 2020.
> [2] Siyao Li, et al. Bailando: 3D Dance Generation by Actor-Critic GPT with Choreographic Memory. In Proceedings of the IEEE Conference on Computer Vision and Pattern Recognition. 2022.
> [3] Jinbo Xing, et al. CodeTalker: Speech-Driven 3D Facial Animation with Discrete Motion Prior. In Proceedings of the IEEE Conference on Computer Vision and Pattern Recognition. 2023.